# High-Throughput Screening of High-Performance Thermoelectric Materials with Gibbs Free Energy and Electronegativity

**DOI:** 10.3390/ma16155399

**Published:** 2023-08-01

**Authors:** Guiying Xu, Jiakai Xin, Hao Deng, Ran Shi, Guangbing Zhang, Ping Zou

**Affiliations:** 1Beijing Municipal Key Lab. of Advanced Energy Materials and Technology, School of Materials Science and Engineering, University of Science and Technology Beijing, Beijing 100083, China; 15313135265@163.com (J.X.); dh838864632@163.com (H.D.); shiran18505297566@163.com (R.S.); zhangbin8236@163.com (G.Z.); 2School of Materials Science and Engineering, Guizhou Minzu University, Guiyang 550025, China; zouping0813@163.com

**Keywords:** high-throughput screening, thermoelectric materials, Gibbs free energy, electronegativity, screening criteria for thermoelectric materials

## Abstract

Thermoelectric (TE) materials are an important class of energy materials that can directly convert thermal energy into electrical energy. Screening high-performance thermoelectric materials and improving their TE properties are important goals of TE materials research. Based on the objective relationship among the molar Gibbs free energy (G_m_), the chemical potential, the Fermi level, the electronegativity (X) and the TE property of a material, a new method for screening TE materials with high throughput is proposed. This method requires no experiments and no first principle or Ab initio calculation. It only needs to find or calculate the molar Gibbs free energy and electronegativity of the material. Here, by calculating a variety of typical and atypical TE materials, it is found that the molar Gibbs free energy of Bi_2_Te_3_ and Sb_2_Te_3_ from 298 to 600 K (G_m_ = −130.20~−248.82 kJ/mol) and the electronegativity of Bi_2_Te_3_ and Sb_2_Te_3_ and PbTe (X = 1.80~2.21) can be used as criteria to judge the potential of materials to become high-performance TE materials. For good TE compounds, G_m_ and X are required to meet the corresponding standards at the same time. By taking G_m_ = −130.20~−248.82 kJ/mol and X = 1.80~2.21 as screening criteria for high performance TE materials, it is found that the G_m_ and X of all 15 typical TE materials and 9 widely studied TE materials meet the requirement very well, except for the X of Mg_2_Si, and 64 pure substances are screened as potential TE materials from 102 atypical TE materials. In addition, with reference to their electronegativity, 44 pure substances are selected directly from a thermochemical data book as potential high-performance TE materials. A particular finding is that several carbides, such as Be_2_C, CaC_2_, BaC_2_, SmC_2_, TaC and NbC, may have certain TE properties. Because the G_m_ and X of pure substances can be easily found in thermochemical data books and calculated using the X of pure elements, respectively, the G_m_ and X of materials can be used as good high-throughput screening criteria for predicting TE properties.

## 1. Introduction

Thermoelectric (TE) materials have received widespread attention around the world due to their ability to convert heat and electricity directly to each other. Improving their thermoelectric conversion efficiency or finding materials with high thermoelectric properties is a very important goal of thermoelectric material research. In principle, the thermoelectric figure of merit Z = ɑ^2^σ/K or ZT = ɑ^2^σT/K (ɑ, σ, K and T are the Seebeck coefficient, electrical conductivity, thermal conductivity and absolute temperature of the material, respectively) is not only a parameter used to evaluate the performance of a TE material but also a theoretical basis for exploring high-performance TE materials. However, since the ZT of thermoelectric materials usually varies significantly with carrier concentration and temperature, thermoelectric materials with an unknown optimum doping concentration and the maximum figure of merit can only be evaluated by preparing numerous samples with different doping concentrations to measure and analyze the parameters over a wide temperature range. Obviously, the time from material composition design, weighing, synthesis and sintering to thermoelectric property testing and a performance analysis will take as little as three months or as much as one year. Therefore, it is difficult to use Z or ZT values to quickly analyze and judge a large number of unknown materials one by one. With the implementation of genetic engineering projects in recent years, the screening of thermoelectric materials using high-throughput calculations has received widespread attention worldwide. The mainstream high-throughput screening methods are theoretical calculations based on density functional theory and the Boltzmann equation, which establishes the relationship between the lattice structure and the thermoelectric transport coefficient of a material, and uses descriptors such as low thermal conductivity, thermoelectric superiority or the power factor to characterize the thermoelectric properties of the material. For example, elastic properties are used to efficiently evaluate the intensity of anharmonicity and lattice thermal conductivity for the high-throughput and efficient screening of thermoelectric materials with low lattice thermal conductivity [1]. But the high-throughput calculations and screening of high-performance thermoelectric materials also face two important difficulties: (1) precise calculations of the electrical and thermal properties of the materials are difficult and time-consuming, and (2) the existing high-throughput methods for evaluating the electrical and thermal properties of the materials have limitations.

Therefore, there is an urgent need for a simple and effective method to make a preliminary determination of the level of thermoelectric properties of a material or a criterion to determine its potential to become a high-performance thermoelectric material. Earlier, based on the thermoelectric figure of merit (Z) proportional to the previously derived material parameter β (see Equations (1) and (2)), Ioffe proposed a method for finding high-performance TE materials using the β value [2].
(1)$$\mathrm{ZT}=\frac{{\left[\xi -(s+\frac{5}{2})\right]}^{2}}{{\left(\beta \exp\xi \right)}^{-1}+(s+\frac{5}{2})}$$
(2)$$\beta \;=\;5.745\times 10^{-6}\frac{\mu_{c}}{K_{L}}{\left(\frac{m^{*}}{m}\right)}^{3/2}T^{5/2}$$
where ξ, s, μ_c_, K_L_, m*, and m are the reduced Fermi level, scattering factor, carrier mobility, lattice thermal conductivity, effective mass, and the mass of the free electron, respectively. In Formula (2), the effective mass m*, lattice thermal conductivity K_L_ and carrier mobility μ_c_ are generally weakly dependent on the carrier concentration, so the β parameter can be used to initially determine the thermoelectric properties of a material, even for samples that are not optimally doped. Accordingly, Ioffe believes that an effective way to find thermoelectric materials is to first screen the materials for lattice thermal conductivity and then make a further determination by measuring the (m*)^3/2^μ_c_ value of the materials. It is clear that this method avoids the requirement of the optimal doping of the sample, and it is much simpler than the method that uses the original formula of the thermoelectric figure of merit Z or ZT. Nevertheless, the determination of the β parameter involves the measurement of carrier mobility (μ_c_) and effective mass (m*), both of which are more complex to measure than the Seebeck coefficient (ɑ) and electrical conductivity (σ), thus limiting the practical application of this approach. It is also essential to note that, while the variation in the β parameter with carrier concentration is much less pronounced than the thermoelectric figure of merit, it is not a constant.

Based on many years of practice, a number of useful laws, for example, heavy atomic mass [3], a large Fermi surface complexity factor [4], multiple energy valley degeneracies [5,6] or a complex Fermi surface structure [7], the appropriate carrier concentration [8], resonance energy levels [9,10], the energy-filtering effect [11], strong phonon scattering [12], strong anharmonic effects [13] and Phonon Glass–Electron Crystal (PGEC) properties [14], have been gradually obtained that summarize the physical properties of a good thermoelectric material. Among them, the use of materials with a high average relative atomic mass to improve m*/K_L_ and, thus, thermoelectric properties as criteria for selecting thermoelectric materials was first proposed by Goldsmid [15]. This rule was supplemented in 1995 by Slack et al., who noted the relationship of the electronegativity (X) of compounds with mobility, the effective mass and the forbidden band width, and they proposed the use of the electronegativity of compounds as a metric for the first screening of thermoelectric materials. This law can be briefly stated as follows: (1) The greater the sum of the atomic numbers of a compound, the larger the cell size, and the lower the thermal conductivity in general. (2) The smaller the electronegativity of a compound, the larger the product of the effective mass and mobility generally [16]. While the laws summarized by Goldsmid and Slack are useful for a preliminary judgment of element choices for thermoelectric materials, they do not provide sufficient insight into the effects of the elements in the periodic table on the thermoelectric properties of materials. The other parameters listed above suffer from similar problems as β. They all require extensive experimental measurements or calculations to be obtained. They cannot be used as simple, fast and effective criteria to judge whether a compound or alloy has a high thermoelectric performance or thermoelectric figure of merit.

Obviously, the ideal way to find promising thermoelectric materials in a wide variety of materials is to make preliminary judgments based on the periodic table of the elements and the known basic physical properties of the elements. Moreover, it is a fact that there is a wealth of molar Gibbs free energy (G_m_) data or thermochemical data of pure substances and convenient calculation methods for molar Gibbs free energy. In this paper, firstly, the rationality of using molar Gibbs free energy to evaluate the thermoelectric properties of materials is described. Then, the molar Gibbs free energy of a series of typical and atypical pure compound thermoelectric semiconductor materials is shown, the electronegativity (X) of the corresponding materials is calculated, and the change rule is analyzed. A new method using the molar Gibbs free energy and electronegativity of pure compounds as a fast and high-throughput preliminary screening method for thermoelectric materials is discussed.

## 2. Basal Principle

### 2.1. Fermi Level as a Criterion for High-Performance Thermoelectric Materials

This paper assumes that there is only one type of carrier that obeys the Fermi–Dirac statistical distribution, the isoenergy surface is spherical, the energy band is parabolic, the relaxation time approximation can be used to describe the scattering process in the crystal, and the contribution of the drag effect can be neglected; furthermore, the Fermi level E_F_ is considered an independent variable. The relationship between the reduced Fermi level ξ (ξ = E_F_/k_B_T), the Seebeck coefficient (ɑ) and the conductivity (σ) of a material under different degenerate conditions can be seen in Table 1.

As mentioned above, in order to obtain a high thermoelectric figure of merit, Ioffe proposed [15] the use of the β factor to predict the thermoelectric performance of a material. It is believed that the greater the β value, the higher the thermoelectric performance of the material. On this basis, in 1959, Chasmar and Stratton used the Fermi–Dirac statistic to rigorously calculate the dependence of the dimensionless value ZT_max_ on the reduced Fermi level ξ (ξ = E_F_/k_B_T), the β factor and the scattering factor (s). The results are shown in Table 2 [17,18].

It can be found that the ZT_max_ value increases with an increase in the scattering factor (s) and the β value, but it increases with a decreasing optimal reduced Fermi level (ξ_opt_). In addition, in correspondence to the different scattering mechanisms, the optimal reduced Fermi level (ξ_opt_) also has a certain range of variation. When s = −1/2 and 1 ≤ β ≤ 5, the ZT_max_ is between 1.8 and 4.6, and the ξ_opt_ is between −1.0 and −2.4. An inverse relationship can also be seen between ZT_max_ and ξ_opt_.

Furthermore, in 1972, Ure [18,19] used a two-band model and disregarded the effects of multi-valley and non-spherical isoenergetic surfaces to avoid excessive complexity. In the study, Ure considered the effects of lattice thermal conductivity and bipolar diffusion and adopted the elastic constant (1.7 × 10^11^ Nm^−2^), the deformation potential constant D (7 eV) and the effective mass m* (0.014 m) of silicon. For the case where acoustic phonon scattering predominates, Ure used this method to estimate the actual optimal values of thermoelectric materials. The results indicated that the dimensionless optimal upper limit was ZT_max_ ≈ 8, corresponding to the optimal reduced Fermi level ξ_opt_ ≈ −3.0.

As explained in the previous paragraph, it should be emphasized that, although the formulas in Table 1 are based on the above assumptions, the conclusion that the Fermi level and the scattering factor have a decisive effect on the thermoelectric properties of a material is universal. Based on the research results of the related literature on the effects of two kinds of carriers (including holes and electrons), an asymmetric band structure and a dual-band structure (considering a conduction band and a valence band) on thermoelectric properties, it can be confirmed that the Fermi level and scattering factor have a decisive influence on the thermoelectric properties of materials [20,21,22,23]. In addition, the typical high-performance thermoelectric materials known to us, such as bismuth telluride, antimony telluride and lead telluride, have more energy valleys. Therefore, the conclusion that Fermi levels and scattering factors have a decisive effect on the thermoelectric properties of materials is universal.

Summarizing the research results reported above, it should be possible to give a preliminary judgment on the thermoelectric performance of a material based on the Fermi level (E_F_) or reduced Fermi level (ξ). Therefore, the relationship between the Gibbs free energy and the Fermi level (E_F_) of materials is discussed.

### 2.2. The Relationship between Molar Gibbs Free Energy and Fermi Level

According to thermodynamics theory, the Gibbs free energy G(T, P, N) of a material system has an extensive property, where T, P and N are the absolute temperature, pressure and moles of a pure substance. The G(T, P, N) of the system is equal to the product of the number of moles of the substance (N), and the molar Gibbs free energy G_m_(T, P) is equal to the chemical potential (μ) of that substance. This can be expressed as [24,25]
(3)$$\mu =G_{m}\left(T,P\right)=\frac{G(T,P,N)}{N}=H-\mathrm{TS}$$
where G_m_(T, P) (G_m_ for short), H and S are the Gibbs free energy, enthalpy and entropy per mole of a pure substance.

At 0 K, the Fermi level (E_F_) of a pure substance is equal to its chemical potential μ [24], namely,
E_F_ = μ(4)

By substituting Formula (3) into Formula (4), we can obtain the following formula:(5)$$E_{F}{\;=\mu =G}_{m}(T,P)=H-\mathrm{TS}$$

In Formula (5), it appears that the Fermi level can be solved using the molar Gibbs free energy of the material. However, in addition to the fact that the two are equal at absolute temperatures and not necessarily exactly equal at other temperatures, the ground states calculated for the two are also different. E_F_ has a ground state temperature of 0 K. That is, the Fermi energy is defined as the energy of the topmost filled level in the ground state of the N electron system. The ground state is the state of the N electron system at absolute zero. However, by definition, for a homogeneous crystal with a uniform temperature, S is calculated as [26]
(6)$$S(T)={\mathrm{kln}\Omega }_{C}(T=0)+\int_{0}^{T}{(C}_{p}/T)\mathrm{dT}$$
where C_p_(T) and Ω_c_ are the isobaric heat capacity and the thermodynamic probability of the material, respectively. For a perfect crystal, Ω_c_(T = 0) = 1; that is, S(T = 0) = 0.

The enthalpy H(T, P) (H for short) of a pure substance is described entirely by independent internal variables T and P. The state function H(T, P) is determined when the pressure P is constant, except for additional uncertain and arbitrarily selected constants. In other words, for any system, the absolute value of enthalpy (H) cannot be determine. For this reason, different books use different conventions for the zero of H, such as the data cited here stating that the standard enthalpy H of all pure elements in its reference phase is zero at P = 1 bar and T = 298.15 K. Therefore, when using enthalpy values from different sources, we must pay attention to the standard state of the reference phase. For pure material enthalpy, its calculation formula can be expressed as [26]
(7)$$H(T)=H(298.15)+\int_{0}^{T_{t1}}C_{p}(T)\mathrm{dT}+{\Delta H}_{t1}+\int_{T_{t1}}^{T_{t2}}C_{p}(T)\mathrm{dT}+{\Delta H}_{t2}+\cdot \cdot \cdot$$
where H(298.15) is the enthalpy of the formation of pure matter at 1 bar and 298.15 K, C_p_(T) is the temperature function of the heat capacity, and ΔH_t1_ is the enthalpy of the phase transition at temperature T = T_t1_. The corresponding entropy calculation Formula (6) can also be expressed as follows:(8)$$S(T)=S(298.15)+\int_{0}^{T_{t1}}\frac{C_{p}\left(T\right)}{T}\mathrm{dT}+\frac{{\Delta H}_{t1}}{T_{t1}}+\int_{T_{t1}}^{T_{t2}}\frac{C_{p}\left(T\right)}{T}\mathrm{dT}+\frac{{\Delta H}_{t2}}{T_{t2}}+\cdot \cdot \cdot$$
(9)$$S(298.15)=S(T=0)+\int_{0}^{298.15}\mathrm{dH}/T$$
(10)$${S=\mathrm{kln}\Omega }_{C}(T=0)+\int_{0}^{T}{(C}_{p}/T)\mathrm{dT}$$

For a complete crystal, at T = 0 K, S = 0, which is Ω = 1.
(11)$$S(298.15)=\int_{0}^{298.15}\mathrm{dH}/T$$

Then, the molar Gibbs free energy of a pure substance at 1 bar can be calculated as follows:(12)$$G_{m}{(T)=G}_{m}(298.15)+\int_{0}^{T_{t1}}C_{p}{(T)\mathrm{dT}+\Delta H}_{t1}+\int_{T_{t1}}^{T_{t2}}C_{p}{(T)\mathrm{dT}+\Delta H}_{t2}-T(\int_{0}^{T_{t1}}\frac{C_{p}\left(T\right)}{T}\mathrm{dT}+\frac{{\Delta H}_{t1}}{T_{t1}}+\int_{T_{t1}}^{T_{t2}}\frac{C_{p}\left(T\right)}{T}\mathrm{dT}+\frac{{\Delta H}_{t2}}{T_{t2}}+)\cdot \cdot \cdot$$
(13)$$G_{m}(298.15)=H(298.15)-\mathrm{TS}(298.15)$$

In Equation (13), H(298.15) and S(298.15) are the standard enthalpy and the standard entropy of the pure substance, respectively. Therefore, G(T) function values are also involved in the H(T) convention. Thus, the molar Gibbs free energy of the reference phase of an element E at 198.15 K and 1 bar is given by using the following formula:(14)$$G_{m}(298.15)=-\mathrm{TS}(298.15)$$

If there is no phase transition, Equation (12) can be simplified as follows:(15)$$G_{m}\left(T\right)=G_{m}(298.15)+\int_{0}^{T}C_{p}(T)\mathrm{dT}-T\int_{0}^{T}\frac{C_{p}\left(T\right)}{T}\mathrm{dT}$$

Therefore, the chemical potential (μ) or Fermi level (E_F_) cannot be calculated simply from the data of the molar Gibbs free energy. However, because of the inevitable relationship between the two, we can still summarize the inevitable relationship between the change law of the molar Gibbs free energy data of thermoelectric materials and the thermoelectric properties of materials and use it as one of the methods for the high-throughput screening of thermoelectric materials.

In addition, as mentioned above, Slack et al. noted the relationship between the electronegativity and mobility, effective mass and band gap width in compounds, and they proposed using the electronegativity of a compound as a metric for a preliminary screening of thermoelectric materials [16]. To this end, following the example of Bulter and Ginley et al., the electronegativity X of the semiconductor compound A_n_B_m_ is calculated using the geometric mean value of Mulliken electronegativity [27,28]:(16)$$X={{(X}_{A}^{n}X_{B}^{m})}^{1/(n+m)}$$
where X_A_ and X_B_ are the electronegativity of pure elements A and B, respectively.

## 3. Results and Discussion

Except for special emphasis, the molar Gibbs free energy (G_m_) data are selected from Ihsan Barin’s *Thermochemical Data of Pure Substances* [26] or the *Handbook of Inorganic Thermodynamics Data* [29].

### 3.1. Molar Gibbs Free Energy (G_m_) of Pure Elements Listed in the Seebeck and Meissner Sequences

The G_m_s of the substances listed in the Seebeck and Meissner sequences [30] are shown in Table 3. The elements with high Seebeck coefficients listed in both sequences are Bi and Sb. Their G_m_ values are between −13.572 and −38.325 kJ/mol. Moreover, it is found that their G_m_ values decrease with an increase in the temperature (the absolute value increases). If G_m_ = −13.572 and −38.325 kJ/mol are taken as screening criteria, all elements in Table 3, except for element C, may have certain TE properties in the appropriate temperature range, of which the difference is that their optimal working temperature zones are different. At room temperature, only the G_m_ values of Na, U, Sn, Cd and Au meet the requirements. Although the G_m_ values of K, Hg, Pb and Cs at 298.15 K are within the above range, their absolute G_m_ values are larger or comparable to the G_m_ values of Bi or Sb at higher temperatures, so it is judged that these four elements may have better thermoelectric properties at slightly higher temperatures.

### 3.2. Molar Gibbs Free Energy and Electronegativity of 15 Typical TE Materials

Table 4 shows the G_m_ and X of 15 typical TE materials [31,32,33,34,35,36,37,38,39,40,41,42,43,44,45,46,47,48,49,50,51]. It can be seen that, in the range of 298.15~600 K, the optimal operating temperature range of Bi_2_Te_3_ and Sb_2_Te_3_, their G_m_ values are between −130.196 and −248.819 kJ/mol. If these data, or G_m_ = −130.196~−248.819 kJ/mol, are used as the screening criteria, it can be seen that all the above typical materials have good thermoelectric properties in a certain temperature range, indicating that the G_m_ of Bi_2_Te_3_ and Sb_2_Te_3_ at 298.15~600 K is feasible as the basic standard for a preliminary screening of high-thermal-power-factor thermoelectric materials.

Table 5 lists the electronegativity of 15 typical thermoelectric compounds calculated using Equation (16) and the electronegativity data of the elements [52,53,54]. It can be seen that, since the same element has different electronegativity values, the electronegativity of the corresponding compound is not unique. In addition, the electronegativity of Bi_2_Te_3_ and Sb_2_Te_3_ are very close. Considering that PbTe is a typical medium-temperature thermoelectric material, the electronegativity values of Bi_2_Te_3_, Sb_2_Te_3_ and PbTe, that is, X = 1.80~2.21, are used as the criteria for screening thermoelectric materials. It can be seen that all other materials, except for Mg_2_Si, have electronegativity values that meet this requirement, indicating that X = 1.80–2.21 is a suitable criterion.

### 3.3. Molar Gibbs Free Energy and Electronegativity of 102 Atypical TE Materials

The temperature dependences of the G_m_ of 102 atypical pure compounds are obtained from References [26,29]. The electronegativity X of the 102 pure atypical compounds are calculated using Equation (16). They are shown in Table 6. If G_m_ = −130.196~−248.819 kJ/mol is used as the screening criterion for good TE materials, 67 compounds are screened. It can be found that, in addition to Cu_2_S, Cu_2_Te, Ag_2_S, Ag_2_Se, Ag_2_Te, SnTe and PbSe, which have been widely investigated as high-performance TE materials [55,56,57,58,59,60,61,62], Bi_2_S_3_, Sb_2_S_2_, Mn_3_Si, CoSb_2_, MoSi_2_, MnS, MnSe, MnTe_2_, FeS, FeS_2_, FeSe_0.96_, FeTe_0.9_, FeTe_2_, CoS_0.89_, CoS_2_, NiSe_1.05_, NiSe_1.143_, NiSe_1.25_, NiSe_2_, NiTe, NiTe_1.1_, NiS_2_, NiSe_2_, CuS, InSb, GeS, GeSe, SnS, PbS, AgP_2_, AgP_3_, BeS, Be_2_C, Ba_2_C, AlAs, AlP, AlSb, CaC_2_, CaH_2_, CaPb, Ca_2_Pb, CaSi, NaTe, NaTe_3_, NbC, NbSi_2_, InSe, CaSi_2_, Ca_2_Si, CaSn, CaZn, CaZn_2_, CrS, CrSi_2_, GaP, GaSb, GaSe, GaTe, InP, InS, CuO and Cu_2_O may be good TE materials at suitable temperature ranges. If the G_m_ of Bi_2_Te_3_ at a temperature of 298–800 K, or G_m_ = −1.61~−3.36 eV, is used as the standard, it can be found that the other 10 compounds, namely, TiS, MoS_2_, WS_2_, MnS_2_, CoP_3_, CaTe, FeO, NiO, CdO and SnO, may be TE materials.

If X = 1.80~2.21 is used as the screening criterion of high-performance TE materials, 67 compounds are screened out. A comparison of the screening results of the two methods shows that their results are not completely consistent, although most of them are. For compounds that meet the G_m_ screening criteria, the main difference is reflected in alkali metal and alkaline earth metal compounds. These compounds, such as the alkali metal compounds NaTe and NaTe_2_ and the alkaline earth metal compounds CaH_2_, CaPb, Ca_2_Pb, CaSi, Ca_2_Si, CaSn, CaTe, CaZn and CaZn_2_, are less electronegative than the screening criteria. The X value of some transition metal compounds, such as Mn_3_Si, is also lower than the screening criterion. Transition metal oxides or sulfides, such as FeO, CuO, NiO, CdO, SnO and NiS_2_, have a larger X value than the screening criteria due to the high electronegativity of O or S. Therefore, although they have TE properties, they are not very good TE materials. So, a bigger X is not better. If both G_m_ and X are met as screening criteria, a total of 60 pure compounds have the potential to become high-quality TE materials. They are Cu_2_S, Cu_2_Te, Ag_2_S, Ag_2_Se, Ag_2_Te, SnTe, Bi_2_S_3_, Sb_2_S_2_, CoSb_2_, TiS, MoS_2_, MoSi_2_, WS_2_, MnS, MnSe, MnTe_2_, FeS, FeS_2_, FeSe_0.96_, FeTe_0.9_, FeTe_2_, CoS_0.89_, CoS_2_, CoP3, NiS, NiSe_1.05_, NiSe_1.143_, NiSe_1.25_, NiSe_2_, NiTe_1.1_, NiS_2_, NiSe_2_, CuS, InSb, GeS, GeSe, SnS, PbS, AgP_2_, AgP_3_, BeS, Be_2_C, Ba_2_C, AlAs, AlP, AlSb, CaC_2_, NbC, NbSi_2_, InSe, CaSi_2_, CrS, CrSi_2_, GaP, GaSe, GaTe, InP, InS and Cu_2_O.

### 3.4. Molar Gibbs Free Energy (G_m_) and Electronegativity of Some Potential TE Materials

Based on the above G_m_ and X criteria, 44 possible high-performance thermoelectric compounds are screened directly from the pure substance thermochemical data book [29]. Their electronegativity values are calculated according to Formula (16). The results are presented in Table 7. There are several compounds, such as GeS_2_, MgB_4_, Mo_3_Si, OsSe_2_, Pd_4_S, PtBr_2_, PtI_4_ and ReS_2_, whose electronegativity values deviate from the screening criteria.

### 3.5. The Procedure for the High-Throughput Screening of TE Materials with Gibbs Free Energy and Electronegativity

Because the molar Gibbs free energy of a compound is easily found in the thermochemical data book or calculated, and its electronegativity is easily calculated using the geometric mean value of Mulliken electronegativity, the potential of a material as a high-performance thermoelectric material can be easily and quickly determined. In order to facilitate the screening of TE materials using molar Gibbs free energy (G_m_) and electronegativity (X), a schematic diagram of the screening process is shown in Figure 1.

Additionally, one problem should be discussed. From the results of the analysis of the whole paper, the only typical TE compound that cannot meet the requirements of G_m_ = −130.20~−248.82 kJ/mol and X = 1.80~2.21 at the same time is Mg_2_Si. That is, its X does not meet the requirements because the electronegativity of the element Mg is too low. But why can Mg_2_Si become a typical thermoelectric material? The first reason is that the G_m_ of Mg_2_Si meets the requirements. The second reason is that X can meet the requirements by changing its composition, which is also the strategy adopted in the research process of Mg_2_Si TE materials. Therefore, when screening thermoelectric materials, G_m_ data can be used as the main data, supplemented by X data. It is a reasonable improvement strategy to adjust the composition of a TE material so that its X value meets the requirements.

## 4. Conclusions

Screening high-performance thermoelectric materials and improving their thermoelectric properties are important goals of thermoelectric materials research. Based on the objective relationship among the molar Gibbs free energy (G_m_), the chemical potential, the Fermi level, the electronegativity (X) and the TE property of a material, a new method using molar Gibbs free energy (G_m_) and electronegativity (X) for the high-throughput screening of thermoelectric materials is proposed. The molar Gibbs free energy of 15 typical TE materials, 9 widely studied thermoelectric materials and 93 atypical thermoelectric materials were obtained from a thermochemical data book. The electronegativities of the materials above were calculated using the geometric mean value of Mulliken electronegativity. The feasibility of using G_m_ and X as high-throughput screening thermoelectric materials is discussed in detail. The results are described below.

1. Because it is universal that Fermi levels and scattering factors have a decisive effect on the thermoelectric properties of materials, taking the molar Gibbs free energy Gm and electronegativity X as screening criteria for high-performance TE materials is reasonable.

2. The molar Gibbs free energy G_m_s of typical TE materials Bi_2_Te_3_ and Sb_2_Te_3_ range from −130.196 to −248.819 kJ/mol. The electronegativity Xs of Bi_2_Te_3_, Sb_2_Te_3_ and PbTe range from 1.80 to 2.21. If G_m_ = −130.20~−248.82 kJ/mol and X = 1.80~2.21 are used as screening criteria for high-performance TE materials, the G_m_ and X of all of 15 typical TE materials and 9 widely studied thermoelectric materials meet the requirements very well, except for the X of Mg_2_Si. It is indicated that G_m_ = −130.20~−248.82 kJ/mol and X = 1.80~2.21 are suitable criteria for screening high-performance TE materials.

3. For TE materials, such as Mg_2_Si, due to the extremely low electronegativity of the component elements, its X value cannot meet the requirements, but its G_m_ can meet the requirements very well. G_m_ data can be used as the main data, supplemented by X data. It is a reasonable improvement strategy to adjust the composition of a TE material so that its X value meets the requirements.

4. For good TE compounds, if G_m_ and X are required to meet the corresponding standards at the same time, and G_m_ = −130.196~−248.819 kJ/mol and X = 1.80~2.21 are used as screening criteria, 60 pure substances, including 9 widely studied TE materials, are screened as potential TE materials from 102 atypical TE materials.

5. With reference to their electronegativity, 44 pure substances are selected directly from the thermochemical data book as potential high-performance thermoelectric materials. A particular finding is that several carbides, such as Be_2_C, CaC_2_, BaC_2_ and NbC, may have certain TE properties.

6. Compared with G_m_ = −130.196~−248.819 kJ/mol, the elemental elements in the Seebeck or Meissner sequence are not good thermoelectric materials. This is consistent with the actual results.

7. The G_m_ of pure substances can be easily found in thermochemical data books, and the X of compounds can be calculated easily from the X of pure elements, so using G_m_ and X as high-throughput screening criteria for predicting thermoelectric properties is much more convenient than using the TE figure of merit Z or ZT or the Ab initio calculation method. This method requires no experiments and no first principle or Ab initio calculation.

## Figures and Tables

**Figure 1 materials-16-05399-f001:**
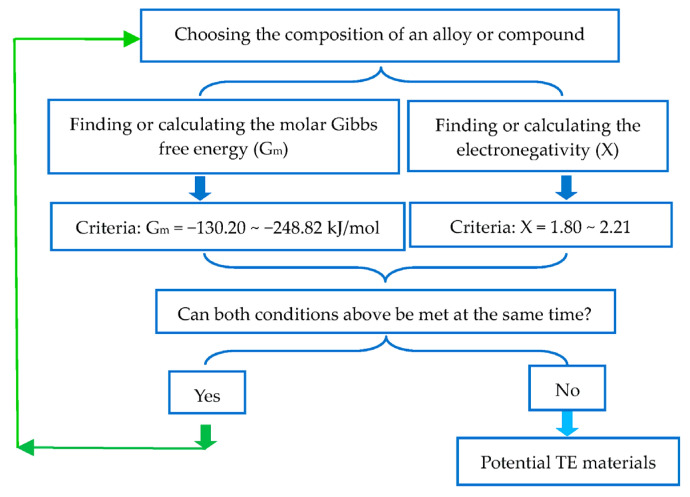
Flowchart for screening high-performance of TE materials using molar Gibbs free energy (G_m_) and electronegativity (X).

**Table 1 materials-16-05399-t001:** The Seebeck coefficients and electrical conductivities under different ξ values.

ξ	Seebeck Coefficient (ɑ)	Electrical Conductivity (σ)
Any ξ	$\alpha_{p,n}=\pm \frac{k_{B}}{e}\left(\frac{(s+\frac{5}{2}{)F}_{s+\frac{3}{2}}\left(\xi \right)}{(s+\frac{3}{2}{)F}_{s+\frac{1}{2}}\left(\xi \right)}-\xi \right)$	$\sigma =\frac{{16\pi e}^{2}\tau_{0}{{(2m}^{*}{)}^{1/2}(k_{B}T)}^{3/2+s}}{{3h}^{3}}(s+\frac{3}{2}{)F}_{s+1/2}\left(\xi \right)$
ξ << 1	$\alpha =\pm \frac{k_{B}}{e}[(s+\frac{5}{2})-\xi ]$	$\sigma =\frac{{16\pi e}^{2}\tau_{0}{{(2m}^{*})}^{1/2}{{(k}_{B}T)}^{3/2+s}}{{3h}^{3}}\Gamma (s+\frac{5}{2})\exp(\xi )$
ξ > 4	$\alpha =\pm \frac{\pi^{2}}{3}\frac{k_{B}}{e}\frac{(s+\frac{3}{2})}{\xi }$	$\sigma =\frac{{16\pi e}^{2}\tau_{0}{{(2m}^{*})}^{1/2}}{{3h}^{3}}E_{F}^{s+3/2}$

**Table 2 materials-16-05399-t002:** Correspondence amongst the optimal reduced Fermi level (ξ_opt_), the scattering factor (s), the material parameter (β) and the figure of merit (ZT_max_).

S	β	0.1	0.2	0.5	1.0	2.0	5.0
−1/2	ZT_max_	0.4	0.6	1	1.8	2.9	4.6
	ξ_opt_	−0.1	−0.1	−0.8	−1.0	−1.7	−2.4
1/2	ZT_max_	0.8	1.1	2.0	2.8	3.8	5.6
	ξ_opt_	0.8	0.35	−0.4	−0.9	−1.6	−2.3
3/2	ZT_max_	1.4	2.0	2.8	3.8	5.0	6.6
	ξ_opt_	1.2	0.5	−0.3	−0.8	−1.5	−2.2

**Table 3 materials-16-05399-t003:** Temperature dependence of the G_m_ of elementary substances listed in Seebeck and Meissner sequences.

Seebeck Sequence	Meissner Sequence	α/ (μV/K)	G_m_/kJ/mol								
			298 K	300 K	400 K	500 K	600 K	700 K	800 K	900 K	1000 K
Bi	Bi	−70	−16.916	−17.021	−23.098	−29.848	−38.325	−48.221	−58.528	−69.185	−80.151
Co	Co	−17.5	−8.957	−9.012	−12.414	−16.483	−21.119	−26.25	−31.888	−37.932	−44.361
Ni	Ni	−18	−8.907	−8.962	−12.371	−16.496	−21.243	−26.557	−32.322	−38.478	−44.99
	K	−12	−19.281	−19.401	−26.798	−35.244	−44.309	−53.879	−63.876	−74.247	−84.952
Pd	Pd	−6	−11.277	−11.347	−15.541	−20.408	−25.824	−31.704	−37.99	−47.973	−51.611
	Na	−4.4	−15.341	−15.437	−21.246	−28.337	−36.044	−44.251	−52.878	−61.867	−71.178
Pt	Pt	−3.3	−12.412	−12.489	−17.061	−22.3	−28.083	−34.326	−40.97	−38.836	−55.299
U			−14.994	−15.088	−20.561	−26.783	−33.648	−41.095	−49.086	−57.601	−66.792
	Hg	−3.4	−22.629	−22.770	−30.795	−39.514	−48.780	−51.584			
	Al	−0.6	−8.430	−8.483	−11.699	−15.565	−19.973	−24.852	−30.150	−35.835	−42.645
	Mg	−0.4	−9.743	−9.803	−13.468	−17.790	−22.661	−28.008	−33.780	−39.937	−47.208
	Pb	−0.1	−19.316	−19.436	−26.340	−33.944	−42.122	−51.588	−61.494	−71.776	−82.389
	Sn	0.1	−15.264	−15.359	−20.911	−27.190	−35.389	−44.143	−53.304	−62.823	−72.658
	Cs	0.2	−25.387	−25.544	−35.244	−45.751	−56.887	−68.542	−80.641	−93.127	−99.241
	Y	2.2	−13.248	−13.330	−18.193	−23.742	−29.851	−36.437	−43.442	−50.824	−58.549
Rh	Rh	2.5	−9.393	−9.452	−13	−17.206	−21.958	−27.18	−32.82	−9.393	−45.199
Zn	Zn	2.9	−12.412	−12.489	−17.055	−22.283	−28.061	−34.393	−42.14	−50.28	−67.574
C			−1.712	−1.722	−2.449	−3.473	−4.789	−6.386	−8.248	−10.358	−12.7
Ag	Ag	1.5/2.4	−12.724	−12.803	−17.471	−22.79	−28.639	−34.939	−41.635	−48.684	−56.057
Au	Au	1.5/2.7	−14.161	−14.249	−19.398	−25.198	−31.525	−38.297	−45.457	−52.962	−60.779
Cu	Cu	2.0/2.6	−9.888	−9.949	−13.654	−17.998	−22.862	−28.17	−33.865	−39.904	−46.255
	W	2.5/1.5	−9.738	−9.798	−13.449	−17.729	−22.519	−27.742	−33.339	−39.271	−45.505
Cd	Cd	2.8	−15.444	−15.54	−21.132	−27.41	−34.322	−42.706	−51.516	−60.698	−70.211
	Mo	5.9	−8.525	−8.578	−11.819	−15.692	−20.084	−24.920	−30.142	−35.707	−41.583
Fe	Fe	16	−8.133	−8.184	−11.317	−15.14	−19.559	−24.513	−29.946	−35.891	−42.302
As			−10.646	−10.712	−14.674	−19.276	−24.4	−29.968	−35.924	−42.225	−48.837
Sb	Sb	35	−13.572	−13.657	−18.609	−24.216	−30.355	−36.949	−43.946	−51.31	−61.128
	Se	1000	−12.599	−12.678	−17.314	−22.738	−29.920	−37.691	−45.966	−54.682	−63.789

**Table 4 materials-16-05399-t004:** Temperature dependence of the G_m_ of typical high-performance thermoelectric materials.

Compounds	G_m_/kJ/mol								
	298 K	300 K	400 K	500 K	600 K	700 K	800 K	900 K	1000 K
Bi_2_Se_3_	−211.206	−212.171	−238.210	−267.142	−298.968	−332.723	−368.406	−406.018	−445.559
Bi_2_Te_3_	−155.271	−155.271	−184.203	−215.064	−248.819	−285.466	−324.043		
Sb_2_Se_3_	−190.954	−190.954	−214.100	−241.103	−270.036	−300.897	−333.687	−369.370	−411.804
Sb_2_Te_3_	−130.196	−130.196	−157.199	−187.096	−219.886	−255.569	−293.182		
GeTe	−75.354	−75.521	−85.337	−96.494	−108.745	−121.928	−135.925	−150.652	
PbTe	−101.263	−101.263	−113.801	−126.338	−140.804	−156.235	−171.666	−188.061	−205.420
MnTe	−135.982	−136.947	−146.591	−158.164	−171.666	−185.167	−200.598	−216.029	−231.459
SnSe	−115.730	−115.730	−125.374	−135.982	−148.520	−161.057	−175.523	−189.989	−205.420
CoSb_3_	−110.908	−110.908	−129.231	−149.484	−171.666	−196.740	−222.779	−250.747	−279.680
Cu_2_Se	−104.157	−104.157	−118.623	−136.947	−156.235	−177.452	−200.598	−223.744	−247.854
Mg_2_Si	−102.228	−102.228	−110.908	−121.516	−133.089	−146.591	−161.057	−176.488	−193.847
MnSi	−91.619	−91.619	−97.406	−104.157	−111.872	−120.552	−130.196	−140.804	−151.413
MnSi_1.7_	−282.573	−281.609	−224.708	−193.847	−174.559	−162.986	−154.306	−149.484	−145.626
Mn_5_Si_3_	−284.502	−283.537	−228.566	−198.669	−180.345	−168.772	−161.057	−156.235	−152.377
FeSi_2_	−97.406	−97.406	−104.157	−112.836	−122.480	−133.089	−145.626	−158.164	

**Table 5 materials-16-05399-t005:** The electronegativity of 15 typical thermoelectric materials.

Element	X_A_	X_B_	A	B	X	Element	X_A_	X_B_	A	B	X
Bi_2_Se_3_	1.67	2.4	2	3	2.08	Mg_2_Si	1.2	1.9	2	1	1.4
Bi_2_Te_3_	1.67	2.1	2	3	1.92		1.2	1.74	2	1	1.36
	1.8	2.1	2	3	1.97		1.2	1.8	2	1	1.37
	2.02	2.1	2	3	2.07		1.2	2.44	2	1	1.52
Sb_2_Se_3_	1.65	2.4	2	3	2.07		1.32	2.44	2	1	1.62
	1.8	2.4	2	3	2.14	MnSi	1.6	1.9	1	1	1.74
	1.82	2.4	2	3	2.15		1.6	1.74	1	1	1.67
	2.05	2.4	2	3	2.25		1.6	1.8	1	1	1.7
Sb_2_Te_3_	1.65	2.1	2	3	1.91		1.6	2.44	1	1	1.98
	1.8	2.1	2	3	1.97	MnSi_1.7_	1.6	1.9	1	1.7	1.78
	1.82	2.1	2	3	1.98		1.6	1.74	1	1.7	1.69
	2.05	2.1	2	3	2.08		1.6	1.8	1	1.7	1.72
GeTe	1.8	2.1	1	1	1.94		1.6	2.44	1	1.7	2.09
	2.01	2.1	1	1	2.05	Mn_5_Si_3_	1.6	1.9	5	3	1.71
	2.02	2.1	1	1	2.06		1.6	1.74	5	3	1.65
PbTe	1.55	2.1	1	1	1.8		1.6	1.8	5	3	1.67
	1.6	2.1	1	1	1.83		1.6	2.44	5	3	1.87
	1.8	2.1	1	1	1.94	FeSi_2_	1.64	1.74	1	2	1.71
	2.33	2.1	1	1	2.21		1.64	1.8	1	2	1.75
MnTe	1.4	2.1	1	1	1.71		1.64	1.9	1	2	1.81
	1.55	2.1	1	1	1.8		1.64	2.44	1	2	2.14
	1.6	2.1	1	1	1.83		1.8	1.74	1	2	1.76
SnSe	1.7	2.4	1	1	2.02		1.8	1.8	1	2	1.8
CoSb_3_	1.7	2.05	1	3	1.96		1.8	1.9	1	2	1.87
Cu_2_Se	1.8	2.4	2	1	1.98		1.8	2.44	1	2	2.2
FeSi_2_	1.83	1.9	1	2	1.88		1.83	1.74	1	2	1.77
	1.83	2.44	1	2	2.22		1.83	1.8	1	2	1.81

**Table 6 materials-16-05399-t006:** The X and the temperature dependence of the G_m_ of atypical thermoelectric compounds.

Compounds	X	G_m_/kJ/mol								
		298 K	300 K	400 K	500 K	600 K	700 K	800 K	900 K	1000 K
Bi_2_S_3_	2.19	−202.527	−203.491	−225.673	−250.747	−277.751	−308.612	−340.438	−374.192	−409.876
Sb_2_S_3_	2.19	−195.776	−196.740	−216.993	−240.139	−265.214	−293.182	−324.043	−359.726	−398.303
Mn_3_Si	1.78	−241.103	−240.139	−193.847	−168.772	−154.306	−145.626	−139.840	−124.409	−135.018
CoSb_2_	1.93	−85.833	−85.833	−99.335	−114.765	−131.160	−149.484	−169.737	−189.989	−211.206
Mg_3_Sb_2_	1.49	−737.776	−732.954	−574.790	−484.135	−427.235	−387.694	−360.691	−340.438	
MgSe	1.70	−1517.022	−1507.378	−1156.331	−948.983	−813.965	−718.488	−649.050	−595.043	−553.573
MgTe	1.59	−1126.435	−1119.684	−868.936	−722.345	−625.904	−558.395	−509.210	−472.562	−443.630
TiS	2.00	−288.360	−289.324	−295.110	−302.826	−311.505	−321.150	−331.758	−342.367	−353.940
TiS_2_	2.15	−430.128	−431.093	−439.772	−450.381	−462.918	−476.420	−490.886	−507.281	−523.676
ZrS_2_	2.11	−600.829	−600.829	−609.509	−620.118	−632.655	−646.157	−660.623	−676.054	−692.449
ZrTe_2_	1.88	−330.794		−345.260		−378.050		−415.662		−458.096
TaS_2_	2.03	−376.121	−376.121	−395.409	−395.409	−407.947	−421.449	−435.915	−451.345	−467.740
MoS_2_	2.15	−295.110	−295.110	−301.861	−311.505	−322.114	−333.687	−346.224	−359.726	−374.192
MoSi_2_	2.12	−137.911	−137.911	−145.626	−155.271	−165.879	−177.452	−190.954	−205.420	−219.886
WS_2_	2.15	−278.715	−278.715	−286.431	−296.075	−306.683	−318.256			
MnS	1.87	−214.100	−214.100	−216.029	−217.957	−219.886	−220.851	−222.779	−276.787	−278.715
MnS_2_	2.15	−253.641	−253.641	−265.214	−277.751	−292.217	−308.612			
MnSe	1.83	−198.669	−198.669	−208.313	−219.886	−232.424	−244.961	−259.427	−273.893	−289.324
MnTe_2_	1.92	−168.772	−168.772	−185.167	−202.527	−221.815	−242.068			
FeS	2.20	−119.587	−119.587	−126.338	−135.982	−146.591	−158.164	−170.701	−184.203	−198.669
FeS_2_	2.17	−187.096	−187.096	−193.847	−201.562	−211.206	−221.815	−233.388	−246.890	−260.392
FeSe_0.96_	2.01	−87.762	−87.762	−95.477	−105.121	−115.730	−127.303	−140.804	−155.271	−170.701
FeTe_0.9_	1.88	−47.256	−47.256	−55.936	−66.545	−77.153	−89.690	−102.228	−115.730	−130.196
FeTe_2_	1.93	−102.228	−102.228	−113.801	−126.338	−141.769	−158.164	−175.523	−193.847	−199.634
CoS_0.89_	2.04	−109.943	−109.943	−115.730	−123.445	−131.160	−139.840	−149.484	−160.093	−170.701
CoS_2_	2.20	−173.594	−173.594	−182.274	−191.918	−202.527	−215.064	−228.566	−243.997	−259.427
CoP_3_	1.99	−309.577	−309.577	−321.150	−335.616	−351.046	−369.370	−388.659	−409.876	−432.057
NiS	2.12	−103.192	−104.157	109.943	−116.694	−125.374	−135.018	−146.591	−158.164	−170.701
NiSe_1.05_	2.09	−97.406	−97.406	−106.085	−115.730	−126.338	−137.911	−151.413	−164.915	−179.381
NiSe_1.143_	2.10	−102.228	−103.192	−111.872	−121.516	−133.089	−145.626	−159.128	−173.594	−188.061
NiSe_1.25_	2.11	−107.050	−107.050	−115.730	−126.338	−137.911	−151.413	−164.915	−180.345	−195.776
NiTe	1.94	−59.794		−68.473		−89.690		−115.730		
NiTe_1.1_	1.95	−82.940	−82.940	−92.584	−103.192	−115.730	−128.267	−141.769	−156.235	−171.666
NiS_2_	2.24	−152.377	−153.342	−161.057	−171.666	−183.238	−195.776	−210.242	−225.673	−241.103
NiSe_2_	2.18	−139.840	−139.840	−151.413	−164.915	−180.345	−196.740	−215.064	−233.388	−253.641
CuS	2.18	−73.295	−73.295	−80.046	−88.726	−98.370	−108.979	−120.552	−132.125	−144.662
Cu_2_S	2.01	−115.730	−115.730	−129.231	−145.626	−162.986	−183.238	−204.456	−226.637	−248.819
Cu_2_Te	1.89	−81.975	−81.975	−97.406	−113.801	−133.089	−154.306	−176.488	−201.562	−226.637
Ag_2_S	2.01	−75.224	−76.189	−91.619	−109.943	−130.196	−151.413	−174.559	−197.705	−222.779
Ag_2_Se	1.98	−82.940	−82.940	−99.335	−120.552	−142.733	−166.843	−191.918	−217.957	−244.961
Ag_2_Te	1.89	−81.975	−81.975	−98.370	−118.623	−141.769	−164.915	−189.989		
ZnSb	1.84	−43.399	−43.399	−53.043	−63.651	−75.224	−86.797	−100.299		
InSb	1.91	−55.936	−55.936	−65.580	−76.189	−88.726	−101.263	−114.765	−135.018	−156.235
GeS	2.12	−95.477	−95.477	−103.192	−111.872	−121.516	−132.125	−143.698	−155.271	−169.737
GeSe	2.08	−92.584	−92.584	−101.263	−110.908	−122.480	−134.053	−146.591	−160.093	−175.523
PbS	2.00	−126.338	−126.338	−135.982	−140.804	−159.128	−172.630	−186.132	−200.598	−216.029
PbSe	1.90	−130.196	−131.160	−141.769	−154.306	−167.808	−182.274	−196.740	−212.171	−228.566
SnS	2.06	−131.160	−131.160	−139.840	−149.484	−160.093	−171.666	−184.203	−197.705	−212.171
SnTe	1.89	−91.619	−91.619	−103.192	−114.765	−128.267	−142.733	−157.199	−172.630	−189.025
AgP_2_	1.99	−69.438	−69.438	−80.046	−91.619	−105.121	−119.587	−135.018	−152.377	
AgP_3_	2.02	−101.263	−101.263	−113.801	−128.267	−145.626	−163.950	−185.167	−207.349	
BeS	1.94	−244.961	−244.961	−249.783	−254.605	−261.356	−268.107	−276.787	−285.466	−294.146
Be_2_C	1.83	−121.516	−121.516	−124.409	−127.303	−132.125	−136.947	−143.698	−150.448	−157.199
BaC_2_	2.11	−101.263	−101.263	−110.908	−122.480	−135.982	−149.484	−164.915	−181.310	−198.669
AlAs	1.82	−134.053	−134.053	−140.804	−149.484	−158.164	−167.808	−178.416	−189.025	−200.598
AlI_3_	2.20	−359.726	−359.726	−380.943	−395.409					
AlP	1.84	−178.416	−758.993	−184.203	−190.954	−197.705	−205.420	−215.064	−223.744	−233.388
Al_2_S_3_	2.04	−758.993	−758.993	−772.495	−788.890	−808.178	−828.431	−851.577	−875.687	−901.726
Al_2_Se_3_	1.99	−613.367	−613.367	−630.726	−650.979	−674.125	−699.200	−726.203	−755.136	−785.032
AlSb	1.82	−69.438	−69.438	−77.153	−85.833	−94.512	−105.121	−115.730	−127.303	−139.840
BaS	1.50	−483.171	−484.135	−491.851	−502.459	−513.068	−524.641	−537.178	−550.680	−564.182
CaC_2_	1.84	−81.011	−81.011	−88.726	−98.370	−109.943	−122.480	−135.982	−151.413	−166.843
CaH_2_	1.64	−189.025	−189.025	−193.847	−200.598	−207.349	−215.064	−223.744	−232.424	−243.032
CaPb	1.53	−144.662	−145.626	−154.306	−163.950	−175.523	−188.061	−200.598	−215.064	−229.530
Ca_2_Pb	1.33	−243.997	−244.961	−256.534	−270.036	−284.502	−300.897	−318.256	−336.580	−355.868
CaS	1.58	−489.922	−489.922	−496.673	−504.388	−513.068	−522.712	−532.356	−543.929	−555.502
CaSi	1.56	−164.915	−164.915	−169.737	−176.488	−183.238	−191.918	−200.598	−210.242	−220.851
CaSe	1.55	−387.694	−388.659	−395.409	−404.089	−413.733	−424.342	−435.915	−447.488	−460.025
NaTe	1.46	−199.634	−199.634	−208.313	−218.922	−230.495				
NaTe_3_	1.75	−165.879	−165.879	−181.310	−198.669	−218.922	−240.139			
NbC	2.06	−149.484	−149.484	−153.342	−159.128	−164.915	−171.666	−179.381	−188.061	−196.740
NbSi_2_	2.16	−146.591	−146.591	−154.306	−163.950	−175.523	−188.061	−200.598	−215.064	−230.495
InSe	1.90	−142.733	−142.733	−151.413	−162.021	−172.630	−185.167	−198.669	−212.171	
CaSi_2_	1.81	−165.879	−165.879	−172.630	−180.345	−189.989	−200.598	−212.171	−224.708	−238.210
Ca_2_Si	1.35	−233.388	−233.388	−243.032	−253.641	−266.178	−280.644	−295.110	−311.505	−328.865
CaSn	1.40	−180.345	−180.345	−188.061	−196.740	−207.349	−217.957	−230.495	−242.068	−255.569
CaTe	1.45	−315.363	−316.328	−324.043	−333.687	−344.295	−354.904	−366.477	−379.014	−391.552
CaZn	1.28	−92.584	−93.548	−100.299	−108.979	−119.587	−130.196			
CaZn_2_	1.40	−124.409	−124.409	−135.982	−149.484	−163.950	−180.345	−197.705	−216.993	
CrS	1.87	−174.559	−174.559	−182.274	−190.954	−199.634	−210.242	−221.815	−234.352	−246.890
CrSi_2_	2.12	−116.694	−116.694	−123.445	−131.160	−140.804	−151.413	−162.986	−175.523	−189.025
GaP	1.95	−115.730	−115.730	−121.516	−128.267	−136.947	−145.626	−155.271	−164.915	−176.488
GaSb	1.93	−64.616	−64.616	−72.331	−81.011	−89.690	−99.335	−109.943	−119.587	
GaSe	2.08	−180.345	−180.345	−188.061	−196.740	−207.349	−217.957	−229.530	−242.068	−255.569
GaTe	1.95	−148.520	−149.484	−158.164	−168.772	−180.345	−192.883	−206.384	−220.851	−235.317
InP	1.93	−106.085	−107.050	−113.801	−121.516	−130.196	−139.840	−150.448	−161.057	−172.630
InS	2.11	−154.306	−154.306	−162.021	−171.666	−181.310	−191.918	−203.491	−216.029	−230.495
Ag_2_O	2.25	−67.509	−67.509	−81.011	−95.477					
MgFe_2_O_4_	2.48	−1474.588		−1490.018		−1528.595		−1577.780		−1635.645
CaTiO_3_	2.33	−1688.687	−1688.687	−1699.296	−1713.762	−1729.193	−1747.517	−1767.769	−1789.951	−1813.097
CaZrO_3_	2.30	−1796.702	−1796.702	−1808.275	−1822.741	−1839.136	−1858.424	−1878.677	−1900.858	−1924.004
SrTiO_3_	2.33	−1705.082	−1705.082	−1717.620	−1733.050	−1750.410	−1770.663	−1791.880	−1815.026	−1839.136
Mn_2_O_3_	2.49	−992.381	−992.381	−1004.919	−1020.349	−1037.709	−1057.961	−1079.178	−1102.324	−1127.399
FeO	2.40	−290.288	−290.288	−297.039	−305.719	−314.399	−325.007	−335.616	−347.189	−359.726
Fe_3_O_4_	2.53	−1162.118	−1162.118	−1179.477	−1200.694	−1226.734	−1255.666	−1287.492	−1323.175	−1360.787
Fe_2_O_3_	2.58	−850.612	−850.612	−861.221	−874.723	−890.153	−909.442	−929.694	−951.876	−976.951
CuO	2.47	−168.772	−168.772	−173.594	−180.345	−187.096	−194.811	−203.491	−213.135	−222.779
Cu_2_O	2.20	−198.669	−198.669	−208.313	−220.851	−234.352	−248.819	−264.249	−280.644	−298.004
NiO	2.47	−250.747	−250.747	−255.569	−263.285	−268.107	−276.787	−285.466	−295.110	−304.755
ZnO	2.29	−363.584	−363.584	−368.406	−375.157	−381.908	−389.623	−398.303	−407.947	−417.591
CdO	2.29	−274.858	−275.822	−281.609	−289.324	−297.039	−306.683	−316.328	−326.936	−338.509
SnO	2.44	−302.826	−302.826	−309.577	−317.292	−325.972	−335.616	−346.224	−356.833	−368.406

**Table 7 materials-16-05399-t007:** The X and the temperature dependence of the G_m_ of some potential TE compounds.

Compounds	X	G_m_/kJ/mol								
		298	300	400	500	600	700	800	900	1000
GeS_2_	2.24	−182.972	−183.134	−192.925	−204.453	−217.428	−231.658	−247.008	−263.375	−280.682
GeSe	2.08	−92.363	−92.509	−101.125	−111.046	−122.039	−133.951	−146.673	−160.125	
GeSe_2_	2.18	−146.525	−146.733	−159.119	−173.371	−189.166	−206.292	−224.597	−243.967	−264.315
InTe	1.94	−103.476	−103.672	−114.999	−127.581	−141.202	−155.719	−171.031	−187.063	
In_2_Te	1.94	−125.861	−126.148	−142.706	−161.059	−180.918	−202.091			
IrS_2_	2.13	−153.634	−153.762	−161.728	−171.474	−182.705	−195.21	−208.836	−223.457	−239.01
MgB_4_	1.62	−120.499	−120.595	−126.939	−135.278	−145.367	−157.024	−170.113	−184.525	−200.173
Mn_7_C_3_	1.83	−180.432	−180.875	−208.533	−242.455	−281.592	−325.235	−372.878	−424.138	−478.719
Mn_4_N	1.81	−171.263	−171.528	−187.866	−207.711	−230.506				
MnP	1.83	−132.428	−132.549	−139.843	−148.441	158.131	−168.751	−180.179	−192.23	−205.1
MnTe_2_	1.92	−168.757	−169.026	−184.752	−202.457	−221.747	−242.362			
Mo_2_N	1.97	−100.425	−100.524	−107.912	−117.106	−127.873	−140.024	−153.411	−167.92	−183.458
MoSi_2_	2.12	−138.21	−138.331	−145.876	−155.167	−165.905	−177.882	−190.944	−204.972	−219.876
Mo_3_Si	1.78	−133.356	−133.554	−145.65	−160.148	−176.604	−194.726	−214.306	−235.189	−257.255
NbC_0.702_	1.99	−126.67	−126.729	−130.466	−135.13	−140.577	−146.704	−153.431	−160.699	−168.459
NbC_0.825_	2.02	−134.083	−134.144	−138.012	−142.865	−148.549	−154.954	−161.993	−169.602	−177.729
NbSi_2_	2.16	−146.353	−146.482	−154.5	−164.221	−175.344	−187.665	−201.037	−215.349	−230.514
Ni_3_Sn	1.84	−132.871	−133.115	−147.837	−165.181	−184.701	−206.103	−229.177	−253.766	
Ni_3_Sn_2_	1.86	−208.67	−208.991	−228.08	−250.01	−274.284	−300.573	−328.641	−358.316	−389.461
OsP_2_	2.07	−176.748	−176.9	−186.225	−197.386	−210.04	−223.964	−238.996	−255.017	−271.933
OsSe_2_	2.26	−144.406	−144.558	−153.754	−164.629	−176.842	−190.157	−204.405	−219.456	−235.21
PdI_2_	2.04	−116.886	−117.219	−136.408	−157.551	−180.296	−204.414	−229.74		
PdS_2_	2.04	−104.438	−104.601	−114.449	−126.078	−139.192	−153.58	−169.089	−185.602	−203.028
Pd_4_S	2.09	−122.901	−123.236	−143.131	−166.052	−191.482	−219.081	−248.607	−279.88	−312.759
PtBr_2_	2.24	−116.346	−116.445	−122.991	−131.511	−141.651	−153.177	−165.924		
PtI_4_	2.24	−126.692	−127.027	−147.125	−170.65	−197.119	−226.212			
PtS_2_	2.08	−132.725	−132.864	−141.394	−151.706	−163.501	−176.571	−190.762	−205.958	−222.065
ReS_2_	2.24	−196.745	−196.858	−203.987	−212.896	−223.29	−234.958	−247.748	−261.542	−276.248
ReSi_2_	2.20	−112.454	−112.591	−121.029	−131.188	−142.771	−155.57	−169.433	−184.245	−199.912
Re_2_Te_5_	2.01	−169.237	−169.704	−197.679	−230.246	−266.6	−306.209	−348.694	−393.774	−441.228
RuSe_2_	2.01	−185.827	−185.979	−195.29	−206.502	−219.273	−233.369	−248.618	−264.893	−282.093
Ta_2_Si	1.92	−156.956	−157.151	−168.802	−182.292	−197.3	−213.601	−231.031	−249.465	−268.806
TaSi_2_	2.16	−135.905	−136.01	−142.703	−151.174	−161.121	−172.329	−184.638	−197.924	−212.09
TiSi	2.04	−144.299	−144.39	−150.02	−156.89	−164.797	−173.598	−183.187	−193.483	−204.421
TiSi_2_	2.16	−152.519	−152.632	−159.805	−168.782	−179.278	−191.085	−204.051	−218.057	−233.01
USi_2_	2.02	−154.154	−154.306	−163.677	−175.054	−188.12	−202.635	−218.415	−235.315	−253.222
VSi_2_	2.13	−168.213	−168.323	−175.267	−183.976	−194.162	−205.616	−218.183	−231.744	−246.204
WSi_2_	2.12	−112.088	−112.207	−119.641	−128.798	−139.381	−151.181	−164.054	−177.858	−192.528
ZrSi	1.91	−172.148	−172.256	−178.783	−186.49	−195.17	−204.68	−214.918	−225.803	−237.276
ZrSi_2_	2.07	−180.742	−180.874	−189.059	−198.947	−210.238	−222.727	−236.266	−250.746	−266.078
SmC_2_	1.94	−122.441	−122.604	−132.522	−144.333	−157.717	−172.445	−188.346	−205.292	−223.18
TaC	2.06	−156.734	−156.812	−161.656	−167.544	−174.326	−181.884	−190.128	−198.986	−208.402
ZnSe	2.00	−179.949	−180.08	−187.926	−197.097	−207.337	−218.474	−230.388	−242.988	−256.205
ZnTe	1.87	−142.447	−142.591	−151.162	−161.036	−171.984	−183.858	−196.552	−209.987	−224.099

## Data Availability

Not applicable.

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
