# Peer review of "High-Throughput Screening of High-Performance Thermoelectric Materials with Gibbs Free Energy and Electronegativity"

_materials, 2023, doi:10.3390/ma16155399_

Round 1
Reviewer 1 Report
The authors suggested a novel approach in their article titled "High-throughput Screening of High-Performance Thermoelectric Materials with Gibbs Free Energy and Electronegativity" that uses the molar Gibbs free energy and electronegativity of pure compounds as a quick and high-throughput preliminary screening method for thermoelectric materials.
The paper has been written and organized well. However, some problems need to be resolved before publication.
1- Explain the reasoning behind the assumptions made in line 117. Please elaborate.
2- In line 65, the equation is not standard in format with other equations. Please type all the equations in MathType and standardize them.
3- The procedure and the content are unclear. Give a flow chart as further explanation.
4- Please be specific in your abstract and conclusion about your novelties.
Author Response
Dear Reviewers:
We thank the learned reviewer for the extensive revision of the document and formaking helpful suggestions and criticism. But we believe that all those comments areposturing criticism which is quite answerable in a justified manner. So, we have revised thepaper and answered the questions asked by the honorable reviewer.
Reviewer #1:“The authors suggested a novel approach in their article titled "High-throughput Screening of High-Performance Thermoelectric Materials with Gibbs Free Energy and Electronegativity" that uses the molar Gibbs free energy and electronegativity of pure compounds as a quick and high-throughput preliminary screening method for thermoelectric materials.
The paper has been written and organized well. However, some problems need to be resolved before publication.”
Our reply: We thank the reviewer for the revision of the manuscript and providing suggestion and comments. We have revised the manuscript, in accordance with the following suggestions of the reviewer. Following are the answers to the Reviewer #1’s comments and criticism.
Reviewer #1 Comment 1:Explain the reasoning behind the assumptions made in line 117. Please elaborate.
Our reply: This paper assumes that there is only one type of carrier that obeys the Fermi-Dirac statistical distribution, the isoenergy surface is spherical, the energy band is parabolic, the relaxation time approximation can be used to describe the scattering process in the crystal, the contribution of the drag effect can be neglected, and the Fermi level EF is considered as an independent variable.
The reason for assuming that “there is only one type of carrier that obeys the Fermi-Dirac statistical distribution” is that only the majority carriers are considered, and the minority carriers are not considered.In addition, except under non-degenerate or strongly degenerate conditions, the carriers usually obey the Fermi-Dirac statistical distribution.
The reason for assuming that “the isoenergy surface is spherical” is that the case where the carrier's equienergy surface is non-spherical is not considered.
The reason for assuming that “the energy band is parabolic” is thatnon-parabolic band structure is not considered.
The reason for assuming that “the relaxation time approximation can be used to describe the scattering process in the crystal,” is that relaxation time approximation is one of the necessary conditions for solving the Boltzmann equation for thermoelectric materials.
The reason for assuming that “the contribution of the drag effect can be neglected” is thatThe drag effect is a low temperature effect, and above room temperature its effect is so small that it is negligible..
The reason for assuming that “Fermi level EF is considered as an independent variable” is to emphasize the relationship between the thermoelectric properties of materials and Fermi levels..
Reviewer #1 Comment 2: In line 65, the equation is not standard in format with other equations. Please type all the equations in MathType and standardize them.
Our reply: It has been revised as shown in the paper.
Reviewer #1 Comment 3: The procedure and the content are unclear. Give a flow chart as further explanation.
Our reply: This is a very good suggestion. We added section 3.5 to the original text. In this section, the process and criteria for rapid screening of thermoelectric materials are given in Fig. 1.
Reviewer #1 Comment 4: Please be specific in your abstract and conclusion about your novelties.
Our reply: The abstract, the conclusion and the text have been revised as suggested.

Reviewer 2 Report
The article presents a study of screening criteria for high-performance thermoelectric materials (TE). In particular, it uses the molar Gibbs free energy and the electronegativity as criteria to determine whether a series of compounds are good TE or not. It is found that a series of substances can meet the standard for thermoelectric performance based on the above criteria and some other may have thermoelectric properties.
The study is interesting and has been properly carried out. The article is well written but, however, has several typos, specially, in the formulas, many of which have incorrectly placed superscripts or subindexes (specially at the beginning) and are displaced. In addition, there are some issues that should be addressed before further consideration.
- Some things need to be more properly explained. For instance, the ranges used in some variables such as Gm or the electronegativity as criterion to determine if a material is a good TE.
- The article assumes several approximations (section 2.1) that are used to determine the screening criteria. The authors should at least explain the limitations of such approximations and to which extent the results can be trusted. They should also explain in some more detail how different criteria are derived from such approximations.
- The results are not always consistent with the expected outcomes and the criteria do not seem to work very well in different cases. The authors should explain why such inconsistencies and lack of correct predictions are arising and if the results could be improved.
The quality of English seems to be fine.
Author Response
Dear Reviewers:
We thank the learned reviewer for the extensive revision of the document and for making helpful suggestions and criticism. But we believe that all those comments are posturing criticism which is quite answerable in a justified manner. So, we have revised the paper and answered the questions asked by the honorable reviewer.
Reviewer #2: The article presents a study of screening criteria for high-performance thermoelectric materials (TE). In particular, it uses the molar Gibbs free energy and the electronegativity as criteria to determine whether a series of compounds are good TE or not. It is found that a series of substances can meet the standard for thermoelectric performance based on the above criteria and some other may have thermoelectric properties.
Our reply: We thank the reviewer for the revision of the manuscript and providing suggestion and comments. We have revised the manuscript, in accordance with the following suggestions of the reviewer. Following are the answers to the Reviewer #2’s comments and criticism.
Reviewer #2 Comment 1: The study is interesting and has been properly carried out. The article is well written but, however, has several typos, specially, in the formulas, many of which have incorrectly placed superscripts or subindexes (specially at the beginning) and are displaced. In addition, there are some issues that should be addressed before further consideration.
Our reply: The sentence “Earlier, Ioffe proposed [2] a method based on the thermoelectric figure of merit (Z) proportional to the previously derived material parameter β, see equation (1), i.e.” has been revised. Similar other problems are revised too.
Reviewer #2 Comment 2: Some things need to be more properly explained. For instance, the ranges used in some variables such as Gm or the electronegativity as criterion to determine if a material is a good TE.
Our reply: Gm = -130.20 ~ -248.82 kJ/mol and X = 1.80 ~ 2.21 are selected as the criteria for high performance thermoelectric compounds because the materials with high thermoelectric properties reported so far except Mg2Si can meet both requirements. For Mg2Si, due to the extremely low electronegativity of Mg, its X value can not meet the requirements, but its Gm can meet the requirements very well. According to formula (16), the X value can be adjusted by changing the element composition, which is also one of the main ways to improve the thermoelectric properties of Mg2Si materials. To this end, recommendations for methods of screening such materials have been added to section 3.5. Therefore, it is reasonable to select Gm = -130.20 ~ -248.82 kJ/mol and X = 1.80 ~ 2.21 as the criteria for screening thermoelectric materials.
Reviewer #2 Comment 3: The article assumes several approximations (section 2.1) that are used to determine the screening criteria. The authors should at least explain the limitations of such approximations and to which extent the results can be trusted. They should also explain in some more detail how different criteria are derived from such approximations.
Our reply: It should be emphasized that although the formulas in table 1 is based on the above assumptions (section 2.1), the conclusion that the Fermi level and the scattering factor have a decisive effect on the thermoelectric properties of the material is universal. Based on the research results of related literature on the influence of two kinds of carriers including hole and rlrctron, asymmetric band structure, dual band structure considering conduction band and valence band on thermoelectric properties, it can be confirmed that Fermi level and scattering factor have a decisive influence on the thermoelectric properties of materials [21-25]. In addition, the typical high performance thermoelectric materials known to us, such as bismuth telluride, antimony telluride and lead telluride, have more energy valleys. Therefore, the conclusion that Fermi levels and scattering factors have a decisive effect on the thermoelectric properties of materials is universal. Therefore, taking Gm = -130.20 ~ -248.82 kJ/mol and X = 1.80 ~ 2.21 as screening criteria for high performance TE materials is reasonable.
Reviewer #2 Comment 4: The results are not always consistent with the expected outcomes and the criteria do not seem to work very well in different cases. The authors should explain why such inconsistencies and lack of correct predictions are arising and if the results could be improved.
Our reply: From the results of the analysis of the whole paper, the only typical thermoelectric compound that cannot meet the requirements of Gm = -130.20 ~ -248.82 kJ/mol and X = 1.80 ~ 2.21 at the same time is Mg2Si. That is, its X does not meet the requirement. The reason is that the electronegativity of the alkaline earth metal element Mg is too low. But why can Mg2Si become a typical thermoelectric material? The first reason is that Gm of Mg2Si meets the requirements. The second reason is that X can meet the requirements by changing its composition, which is also the strategy adopted in the research process of Mg2Si materials. Therefore, when screening thermoelectric materials, Gm data can be the main, supplemented by X data. It is a reasonable improvement measure to adjust the composition of the material so that its X value meets the requirements. This paragraph has been added to Section 3.5 of the paper.